# β-COP Suppresses the Surface Expression of the TREK2

**DOI:** 10.3390/cells12111500

**Published:** 2023-05-29

**Authors:** Seong-Seop Kim, Jimin Park, Eunju Kim, Eun Mi Hwang, Jae-Yong Park

**Affiliations:** 1School of Biosystems and Biomedical Sciences, College of Health Sciences, Korea University, Seoul 02841, Republic of Korea; mykss81@naver.com (S.-S.K.); park971225@naver.com (J.P.); 2BK21FOUR R&E Center for Learning Health Systems, Korea University, Seoul 02841, Republic of Korea; 3Brain Science Institute (BSI), Korea Institute of Science and Technology (KIST), Seoul 02792, Republic of Korea; trek1747@gmail.com; 4ASTRION, Inc., Seoul 02842, Republic of Korea

**Keywords:** β-COP, protein-protein interaction, TREK family, TREK1, TREK2, TRAAK

## Abstract

K2P channels, also known as two-pore domain K^+^ channels, play a crucial role in maintaining the cell membrane potential and contributing to potassium homeostasis due to their leaky nature. The TREK, or tandem of pore domains in a weak inward rectifying K^+^ channel (TWIK)-related K^+^ channel, subfamily within the K2P family consists of mechanical channels regulated by various stimuli and binding proteins. Although TREK1 and TREK2 within the TREK subfamily share many similarities, β-COP, which was previously known to bind to TREK1, exhibits a distinct binding pattern to other members of the TREK subfamily, including TREK2 and the TRAAK (TWIK-related acid-arachidonic activated K^+^ channel). In contrast to TREK1, β-COP binds to the C-terminus of TREK2 and reduces its cell surface expression but does not bind to TRAAK. Furthermore, β-COP cannot bind to TREK2 mutants with deletions or point mutations in the C-terminus and does not affect the surface expression of these TREK2 mutants. These results emphasize the unique role of β-COP in regulating the surface expression of the TREK family.

## 1. Introduction

The cell membrane has a constant membrane potential maintained by various intra-cellular ions [1]. Membrane potential is essential for maintaining the function of various protein machinery and transmitting signals between cells [2]. If the membrane potential is not properly maintained, cells can enter an overexcited state, leading to diseases such as epilepsy [3]. The cell membrane potential is highly sensitive to the extracellular potassium ion concentration, and the membrane potential changes sensitively depending on the potassium concentration [4]. Two pore domain K^+^ (K2P) channels have leaky properties and have been reported to mediate cellular potassium homeostasis [5]. The tandem of pore domains in a weak inward rectifying K^+^ channel 1 (TWIK1) was discovered in 1996, and by 2003, 15 different K2P channels had been discovered [6,7]. They have been classified into six sub-families based on similarities in their functional properties and sequences. These channels contain four transmembrane domains and two pore domains that constitute the filter [8]. In particular, it is characterized by its long N-terminus and C-terminus extending into the cell membrane. Because of this, function and expression of K2P channels are regulated by binding to various proteins, and its location within the cell changes [9,10]. TWIK-related acid-sensitive K^+^ (TASK) 1 and TASK3 are antagonistically regulated by 14-3-3 and COP1 [11,12,13]. 14-3-3 proteins also alter the kinetics of calcium-dependent enhanced activity of the TWIK-related spinal cord K+ channel (TRESK) by directly binding to the intracellular loop of TRESK [14]. In addition, the TWIK1 channel binds to ADP-ribosylation factor 6 (ARF6) [15], and TWIK-related K^+^ channel 1 (TREK1) is upregulated by A-Kinase Anchoring Protein 150 (AKAP150) and Microtubule-associated Protein 2 (Mtap2) [16,17]. TREK1 is also regulated by β-coat protein (β-COP), a subunit of coatomer protein complex 1 (COP1) [18].

These characteristics of K2P channels are particularly noticeable in the TWIK-related K^+^ channel (TREK) subfamily, which includes TREK1, TWIK-related K^+^ channel 2 (TREK2), and TWIK-related acid-arachidonic activated K^+^ (TRAAK) channels [19]. The TREK subfamily contributes to cellular potassium homeostasis by generating outwardly rectifying currents [20]. Members of this subfamily share unique regulatory features. Notably, TREK2 and TREK1 have 63% identity and 78% homology, high sequence similarity, and similar sequences of up to 50 contiguous amino acids following the fourth transmembrane domain [21]. In addition, it is similarly regulated by physical and chemical stimuli such as cell membrane stretch, intracellular acidification, depolarization, and heat, and its activity is regulated by PUFA, such as arachidonic acid and volatile anesthetics such as Chloroform and Halothane [22]. TREK1 and TREK2 also have the same binding protein, and both are upregulated upon binding to AKAP150 and Mtap2 [17]. Unlike TREK1 and TREK2, which have highly conserved similar structures, TRAAK lacks similarity. It has 45% identity and 69% homology with TREK1, and its sequence is different, especially at important locations [5,21]. TRAAK differs from TREK1 and TREK2 in several regulatory mechanisms [23]. BL1249 and halothane, a volatile anesthetic, activate TREK1 and TREK2 but not TRAAK [24]. However, in the case of Ruthenium red, it affects the functions of TREK2 and TRAAK but does not react with TREK1 [25].

Previous studies have reported the binding of proteins related to the TREK family. Previously, it was found that AKAP150 and Mtap2 upregulate the surface expression of TREK1 and TREK2 through association, exerting a synergistic effect that makes TREK1 more active [16,17]. However, they do not bind TRAAK. Additionally, β-COP, a vesicular protein that mediates transport between the Golgi apparatus and the endoplasmic reticulum by binding to various proteins and lipids [26,27], has been found to be associated with TREK1 [18]. Previous studies have indicated that β-COP binds to the N-terminus of TREK1 and increases its surface expression [18]. Recent research has also suggested that β-COP binds to the TWIK1/TREK1 heterodimeric channel by binding to TREK1 and consequently regulates the passive conductance of adult mouse hippocampal astrocytes [28,29]. However, although it is unclear whether β-COP is associated with TREK2 or TRAAK, which are members of the TREK subfamily, β-COP may upregulate not only TREK1 but also TREK2, similar to AKAP150 and Mtap2 [16,17].

In this study, we performed imaging analyses using a heterologous system to con-firm that TREK1 and TREK2 share a common expression region within cells with β-COP, whereas TRAAK does not. We observed that the co-localization signals of TREK1 and β-COP exhibit a range of expression regions from the cell membrane to the entire cell, whereas the co-localization signals of TREK2 and β-COP are only detected in the Golgi apparatus. Additionally, through biochemical experiments to identify the binding site, we discovered that β-COP binds to a specific RRR amino acid motif in the C-terminal region of TREK2, which is completely different from the binding site of TREK1. This finding led to the discovery that β-COP inhibits the surface expression of TREK2, showing a completely different effect from that observed for TREK1. Our study highlights the fact that even subfamilies with similar sequence specificities can lead to conflicting results for the same protein, contradicting the assumption that similar sequences in TREK1 and TREK2 indicate similar mechanisms. Furthermore, this study emphasizes the importance of β-COP in regulating the surface expression of the TREK subfamily and provides a deeper understanding of the association between binding proteins and their binding regions.

## 2. Materials and Methods

### 2.1. Constructs and Gene Information

Full-length mouse *TREK1* (GenBank accession number NM_010607), *TREK2* (GenBank accession number NM_029911), mouse *TRAAK* (GenBank accession number NM_001403912), and mouse *β-COP* (GenBank accession number NM_033370) cDNA and the derived vectors were constructed using reverse transcription polymerase chain reac- tion (RT-PCR). TREK2ΔN, TREK2ΔC, TREK2ΔC1, TREK2ΔC2, TREK2ΔC3, TREK2ΔC4, and TREK2RRRAAA are mutants generated with the EZchangeTM site-directed mutagenesis kit (Enzynomics, Daejeon, Republic of Korea) using the mouse *TREK2* cDNA as a template. Vectors pDEST-Flag-C and pDEST-EGFP-N were cloned using the Gateway cloning method. DsRed-Mem (Clontech Laboratories, Mountain view, CA, USA) was used to label the cell membranes, and pmScarlet_Giantin_C1 (Addgene, Teddington, UK) was used to label the Golgi apparatus.

### 2.2. Human Embryonic Kidney (HEK) 293T Cell Culture

HEK293T cells were purchased from the Korea Cell Line Bank (Seoul National University, Seoul, Republic of Korea). The cells were cultured after adding 100 units of penicillin-streptomycin per milliliter to Dulbecco’s modified Eagle’s medium (DMEM, Invitrogen) supplemented with 10% fetal bovine serum (Invitrogen, Carisbad, CA, USA). It was maintained at 37 °C and incubated in an environment composed of 95% air and 5% CO_2_. According to the manufacturer’s instructions, all genes were transfected into cells using the Lipofectamine^®^ 3000 reagent (Invitrogen, Carisbad, CA, USA). All genes were used for experiments 24 h after transfection into cells.

### 2.3. Bimolecular Fluorescence Complementation (BiFC) Experiment

Before BiFC analysis, *TREK1*, *TREK2*, *TRAAK*, and all *TREK2* mutants were cloned into pBiFC-VN173 and pBiFC-VC155 vectors. To obtain these results, the cells were transfected with vectors containing individual genes tagged at the N- and C-termini and incubated for 24 h. Next, the coverslips on which the cells were placed were fixed with 4% paraformaldehyde for 20 min at 25 °C, and the nuclei were stained with 4′,6-diamidino-2-phenylindole (DAPI) for 5 min. After the coverslips were mounted with Dako Fluorescence Mounting Medium, Venus fluorescence was observed under a Nikon A1 confocal microscope (Nikon Imaging Japan Inc., Tokyo, Japan).

### 2.4. Immunocytochemistry (ICC) and Quantification of Co-Localization

After transfection, HEK293T cells were fixed in 4% paraformaldehyde for 15 min at 25 °C and permeabilized with phosphate-buffered saline (PBS) supplemented with 0.1% TritonX100 for 5 min. Nonspecific binding of the antibody was prevented by incubation for at least 2 h in blocking buffer (PBS-based 5% donkey serum [GeneTex, Irvine, CA, USA]). After the blocking process, cells were incubated overnight at 4 °C in a blocking buffer with anti-Flag antibody (Sigma-Aldrich, St. Louis, MO, USA; 1:200) added. The day after washing, a secondary antibody conjugated to Dye Light 647 (Jackson Laboratories, Bar Harbor, ME, USA; 1:400) was added and incubated for 1 h. The cells were washed and mounted using Dako Fluorescence Mounting Medium. The samples were observed using a Nikon A1 confocal microscope. Intracellular co-localization of channels and β-COP was determined by the confocal microscope as described previously [18]. To quantify co-localization, Pearson’s correlation coefficients were determined using the NIS-Elements AR, (version 5.01) imaging software of a Nikon A1 confocal microscope. 

### 2.5. Co-Immunoprecipitation

HEK293T cells were washed with PBS and incubated for 2 h at 4 °C with Pierce™ IP Lysis Buffer composed of 0.1% sodium dodecyl sulfate (SDS), 50 mM Tris-Cl, 150 mM NaCl, 1% NP-40, 0.5% sodium deoxycholate, and protease inhibitor cocktail (Tech and Innovation, Chuncheon, Republic of Korea). Upon completion of the lysis process, the lysate was centrifuged at 13,000× *g* for 20 min at 4 °C, and the supernatant was transferred to a new tube. After taking 5% of the total lysate as input, the remainder was incubated overnight at 4 °C on a rocking mixer with antibodies (anti-Flag, Sigma-Aldrich; anti-GFP, Santa Cruz Biotechnology, Dallas, TX, USA) for co-immunoprecipitation. The next day, samples were incubated with Protein G Agarose (Santa Cruz Biotechnology) for 1 h at 4 °C. Protein samples were separated by SDS-PAGE on a 10% gel, and the separated proteins were transferred onto a nitrocellulose membrane. Blots were incubated overnight at 4 °C with anti-GFP antibodies (Santa Cruz Biotechnology, 1:1000), anti-Flag antibodies (Sigma-Aldrich, 1:1000), or anti-RFP antibodies (Abcam, Cambridge, UK, 1:1000), depending on the protein to be detected. The blots were washed three times after 40 min of incubation at 25 °C with horseradish peroxidase-conjugated goat anti-rabbit or goat anti-mouse IgG, and immunoreactivity was detected using chemiluminescence (Amersham Biosciences, Piscataway, NJ, USA).

### 2.6. Biotinylation Assay

HEK293T cells transfected with Flag-TREK2 and GFP-Con or GFP-β-COP or HEK293T cells transfected with Flag-TREK2RRRAAA and GFP-Con or GFP-β-COP were incubated for 5 min on ice 24 h after co-transfection was completed. The medium was then removed, and the cells were washed with PBS. The cell membrane-expressed proteins were biotinylated for 30 min in PBS containing sulfo-NHS-SS-biotin (Pierce, Rockford, IL, USA), at 0 °C. After biotinylation, the cells were washed with quenching buffer (100 mM glycine in PBS). Following that, the cells were lysed with RIPA buffer and incubated with high-capacity NeutrAvidin-Agarose Resin (Thermo Scientific, Waltham, MA, USA) at 4 °C for 24 h. After washing three times with RIPA buffer, western blot analysis was performed by eluting the proteins with SDS sample buffer.

### 2.7. Electrophysiological Recording

To obtain electrophysiological data, transfected HEK293T cells were placed on coverslips 24 h after transfection. Transfected cells were immersed in a standard bath solution containing: 150 mM NaCl, 3 mM KCl, 2 mM CaCl_2_, 1 mM MgCl_2_, 10 mM N-2-Hydroxyethylpiperazine-N’-2-Ethanesulfonic Acid (HEPES), 5.5 mM D-glucose, and 20 mM sucrose (pH 7.4, adjusted with NaOH) to measure currents. A patch pipette was created using a borosilicate glass capillary (Warner Instruments, Washington, DC, USA) and filled with a standard solution containing 150 mM KCl, 1 mM CaCl_2_, 1 mM MgCl_2_, 5 mM EGTA, and 10 mM HEPES (pH 7.2, adjusted with KOH). The pClamp software 10.4 (Axon Instruments, Union City, CA, USA) was used to analyze the currents, and the Digidata 1550 A interface (Axon Instruments, Union City, CA, USA) was used to convert the digital-to-analog signals between the computer and amplifier. Current-voltage (I–V) curves were measured by applying a 1-second ramp pulse (from −150 mV to +50 mV) at a holding potential of −60 mV while continuously perfusing the standard bathing solution into the chamber (RC-25 chamber, Warner Instruments, Washington, DC, USA) at a rate of 1 mL/min. The data were sampled at 5 kHz and filtered at 1 kHz. Electrophysiological experiments were performed at 20–24 °C.

### 2.8. Statistics

Analyses were performed using SigmaPlot 10.0 and pClamp software 10.4. Confocal microscopy images were analyzed using ImageJ, and all data were quantified as mean ± standard error of the mean. Confirmation of significance was evaluated using the Student’s *t*-test (paired *t*-test), and the significance level was specified as follows: ns: not significant, ** *p* < 0.01, *** *p* < 0.001.

## 3. Results

### 3.1. Association with β-COP Differs for Each Member of the TREK Family

In our previous study, we reported that the interaction between β-COP and TREK1 led to an increase in the surface expression of TREK1 [18]. Building on this finding, in the present study, we aimed to investigate the interaction of β-COP with other members of the TREK subfamily. To achieve this, we tagged each TREK subfamily protein, TREK1, TREK2, and TRAAK, with GFP, and β-COP was tagged with mCherry to confirm the specificity of their localization. We observed that TREK1 and β-COP co-localized throughout the cell, particularly near the cell membrane, which is consistent with our previous findings [18,29]. Conversely, the co-localization of TREK2 and β-COP was densely localized in a Golgi-like organelle adjacent to the nucleus. Additionally, we did not observe any co-localization of TRAAK with β-COP (Figure 1A). Analysis of Pearson’s correlation coefficients of the merged images also showed that TREK1 and TREK2 correlated with β-COP, but TRAAK did not (Figure 1B).

A BiFC assay was performed simultaneously to check the direct binding between TREK subfamily members and β-COP and their approximate intracellular binding location. For the binding test, the split-Venus fluorescent protein was labeled individually for each target protein with N- and C-terminal parts. When VN- and VC-tagged proteins bind, the separated Venus fluorescent proteins come close to each other, resulting in clear Venus fluorescence. Similar to previous reports [29], a clear BiFC signal was observed throughout the HEK293T cells co-transfected with VN-β-COP and VC-TREK1. In HEK293T cells co-transfected with VN-β-COP and VC-TREK2, the signals were concentrated in a narrow area near the organelles, similar to the Golgi apparatus. No fluorescence was observed in the cells co-transfected with VN-β-COP or VC-TRAAK. mCh-Con was transfected to visually confirm cell remodeling, and the binding region of the TREK subfamily member protein and β-COP was confirmed (Figure 1C).

To specifically confirm and quantify the direct binding of TREK2 and β-COP, a Co-Immunoprecipitation (Co-IP) assay was performed. For this, each Flag-tagged TREK subfamily member and GFP-tagged β-COP were co-transfected into HEK293T cells, and the cells were lysed, IP with an anti-Flag antibody, and blotted with an anti-GFP antibody. Western blot results showed glycosylated ion channels according to the type of ion channels. Clear bands were observed when Flag-TREK1 and GFP-β-COP were co-transfected, and Flag-TREK2 and GFP-β-COP were co-transfected. However, no bands were observed when Flag-TRAAK and GFP-β-COP were co-transfected (Figure 1D). Therefore, TREK1 binds to β-COP near the cell membrane, TREK2 binds to β-COP near the nucleus in a condensed state, and TRAAK does not bind to β-COP.

### 3.2. β-COP Suppresses the Membrane Expression of TREK2

β-COP is a subunit of the COP1 protein, which binds to various channels and is involved in the transport of substances between the cell membrane, the Golgi apparatus, and the endoplasmic reticulum [12,30,31,32,33]. We first wanted to determine whether β-COP regulates the translocation of TREK2, resulting in the high-density localization of TREK2 to the nucleus. To visually look at the intracellular expression location of the protein, we constructed the GFP-TREK2 vector and the mCh-β-COP vector. When GFP-TREK2 was co-transfected with mCh-Con, green fluorescence was observed throughout the cell, including the cell membrane (Figure 2A, white arrows). Conversely, when GFP-TREK2 was cotransfected with mCh-β-COP, high-density granulation and TREK2 expression were observed (Figure 2A, white arrows).

We used a DsRed-Mem vector as a membrane marker and a giantin vector as a Golgi marker to determine the exact location of TREK2 expression. The Flag-TREK2 vector was co-transfected with GFP-Con or GFP-β-COP and DsRed-Mem or Giantin vectors into HEK293T cells. To visualize Flag-TREK2, the co-transfected cells were treated with an anti-Flag antibody. As a result, it was confirmed that the expression pattern of TREK2 overlapped with the membrane marker when β-COP was not expressed (Figure 2B, up). When β-COP was co-transfected, overlapping fluorescence of TREK2 and the Golgi marker was observed (Figure 2B, low). TREK2 is expressed in the cell membrane under normal conditions. However, in the presence of additional β-COP, the intracellular expression region of TREK2 changes to the Golgi apparatus.

Additionally, we performed a surface biotinylation assay to quantitatively confirm whether β-COP regulates the membrane expression of TREK2. The results revealed that the total expression level of TREK2 was not changed by the addition of β-COP, but the strong signal of biotinylated TREK2 was abolished by β-COP (Figure 2C,D). Moreover, whole-cell currents were measured to confirm the functional changes in TREK2 caused by β-COP. Weak K^+^ currents were observed when GFP-Con and mCh-Con were transfected, and weak K^+^ currents were measured. HEK293T Cells transfected with GFP-TREK2 showed large K^+^ currents, whereas cells cotransfected with mCh-β-COP showed half-poor K^+^ currents (Figure 2E,F).

Taken together, β-COP binds to TREK2 and shifts its expression region from the cell membrane to the Golgi apparatus, resulting in reduced membrane expression of TREK2 and significant attenuation of channel function. This is the opposite experimental result to that of TREK1, in which membrane expression is increased and activity is enhanced by β-COP.

### 3.3. C-Terminus of TREK2 Is a Pivotal Binding Region for β-COP

In terms of protein-protein interactions, we attempted to identify the TREK2 binding site to which β-COP binds. K2P channels communicate with various proteins through their elongated intracellular N- and C-termini. Therefore, we constructed mutants that removed the N- and C-termini of TREK2 (TREK2ΔN, TREK2ΔC) (Figure 3A). After GFP was tagged to each mutant, Co-IP was performed by co-transfection with mCh-β-COP. Unlike TREK1, which binds to β-COP at the N-terminal motif, TREK2 binds to β-COP in the TREK2ΔN (Figure 3B). In contrast, in TREK2ΔC, the binding signal with β-COP was abolished. We obtained the same results for BiFC. The vector fused with the VC based on TREK2ΔN showed a strong Venus signal when co-transfected with VN-β-COP, whereas the VC vector fused with TREK2ΔC did not. The signal was still observed at a high density in the Golgi apparatus. Therefore, TREK2 binds to β-COP through the C-terminus, which is a different binding mechanism from that of TREK1, which binds through the N-terminus [18].

### 3.4. RRR Motif in TREK2 Is Essential for Binding to β-COP

To investigate the difference in functional regulation according to the binding site with β-COP, a more detailed determination of the binding site is required. We divided the C-terminal region of TREK2, to which β-COP bound, into quarters and produced mutants lacking each segment. A schematic of each vector and the missing sequence is shown in Figure 4A.

We aligned TREK1 C-terminus sequences corresponding to each deletion mutant. The similarity between each deleted C-terminus segment and the C-terminus of TREK1 was 86% for C1, 14% for C2, and 16% for C3. No sequence corresponded to that of TREK1 in C4 (Appendix A). In other words, we had primarily assumed the presence of binding sites at C2 and C4 with relatively low similarities. In a previous study on the binding recognition motif of β-COP, β-COP was determined to recognize and bind to conserved charged domains such as KKXX, KDEL, and RXR of target proteins [34,35,36]. Therefore, we focused on the RRR sequence conserved in the C-terminus of TREK2 and identified the C1 and C2 segments containing the RRR sequence as candidates recognized by β-COP. To gain confidence, Co-IP was performed by co-transfecting the C-terminus deletion mutants of TREK1 (TREK1ΔC) and TREK2 (TREK2ΔC) with β-COP. In the Co-IP experiment, TREK2 ΔC and β-COP did not show binding signals (Appendix A); therefore, the C1 segment of TREK2 having the same RRR sequence as TREK1 was excluded, and the C2 segment of TREK2 was selected as a strong suspect associated with β-COP.

In the following experiment, to confirm the binding of each TREK1 C-terminal deletion variant to β-COP, all deletion mutants were tagged with VC and co-transfected with VN-β-COP. When the created vector was combined with β-COP, strong BiFC fluorescence was generated. Consequently, BiFC signals were not observed only in TREK2ΔC2 (Figure 4B). Subsequently, Co-IP was performed. GFP-tagged deletion mutants and mCh-β-COP were co-transfected, followed by immunoprecipitation using an anti-GFP antibody. Resultantly, when blotting with an anti-mCh antibody, no blotting band was identified in TREK2ΔC2 (Figure 4C).

Based on the above results, we explored TREK2 binding motifs recognized by β-COP. A mutant was prepared by substituting AAA for RRR in the TREK2 C2 segment, which was expected to bind to β-COP. A Co-IP assay was performed to validate binding with β-COP. The strong IP band observed in the TREK2 wild-type disappeared in TREK2RRRAAA (Figure 4D), and the BiFC test revealed the same results (Figure 4E).

Taken together, β-COP binds through the RRR motif of TREK2. In particular, this β-COP recognizes and binds to the second RRR motif that is not complementary to TREK1 among the two RRR motifs present in TREK2. Through this, the expression of TREK2 is regulated in the opposite direction to that of TREK1.

### 3.5. β-COP Does Not Affect the Membrane Expression of the TREK2RRRAAA Mutant

Before concluding, we tested whether TREK2RRRAAA reduced membrane expression due to β-COP binding to ensure that β-COP recognized the RRR motif of TREK2 as the main target. In the co-localization image of the constructed GFP-TREK2RRRAAA and mCh-β-COP, the overexpression of β-COP did not decrease the membrane expression of TREK2. In addition, unlike wild-type TREK2, TREK2RRRAAA was still expressed in the cell membrane despite co-transfection with β-COP (Figure 5A). In the biotinylation results to quantitatively confirm cell membrane expression, no decrease was observed in the membrane expression of TREK2RRRAAA by β-COP (Figure 5B,C). In the results of measuring the whole cell current to confirm the function of TREK2RRRAAA as a channel, the large current in HEK293T cells co-transfected with GFP-TREK2RRRAAA and mCh-Con was not reduced by mCh-β-COP (Figure 5D,E). Taken together, β-COP interacts with TREK2 by targeting the RRR motif at the C-terminus of TREK2, and this motif is key for β-COP to regulate TREK2 function.

## 4. Discussion

TREK1 and TREK2 in the TREK family are structurally and functionally similar. The reported binding proteins, Mtap2 and AKAP150, do not bind TRAAK but associate with TREK1 and TREK2, thereby increasing their expression [16,17]. However, to our knowledge, this study is the first to discuss β-COP, another protein associated with the TREK family. In this study, β-COP, a vesicle protein [26], was determined to function dif-ferently from each member of the TREK family, despite their similarities in structure, func-tion, and amino acid sequences. We first showed that TREK1 and TREK2, but not TRAAK, bind to β-COP (Figure 1B,C). In this simple experiment, differences were observed for the first time in the expression of TREK1 and TREK2, owing to the presence of β-COP. In the heterologous system, overexpressed TREK1 was co-expressed with β-COP in the cell membrane, whereas TREK2 is co-expressed with β-COP in the Golgi apparatus (Figure 1B). Under normal conditions, TREK2, similar to TREK1, was expressed on the cell membrane (Figure 2A,B). However, when β-COP was administered, the location of TREK2 expression shifted from the membrane to the Golgi apparatus (Figure 2B), and the membrane expression of TREK2 decreased, resulting in a significant decrease in channel function (Figure 2E,F). β-COP recognizes the RRR amino acid motif of TREK2, which is not shared with TREK1, among the two RRR motifs present in the C-terminal region of TREK2 (Figure 4D,E). Therefore, in mutants in which this sequence is substituted with alanine amino acids, no significant change by β-COP is observed (Figure 5).

It has been reported that β-COP is known to recognize and bind highly conserved sequences (di-lysine (KXK), di-arginine (RXR), and KDEL [34,35]). We also found that β-COP binds to an RRR sequence present in the C2 region of TREK2, which is critical for binding with β-COP, as shown in Figure 4. However, we also found in our Co-IP experiments that TREK2ΔC4 has weaker bands compared to other deletion mutants of TREK2 (Figure 4C). Therefore, it is plausible that β-COP also binds to TREK2 by recognizing a non-conserved sequence in the C4 region of TREK2. Future studies for the detailed binding site of β-COP within the C4 region of TREK2 will be required. The present study demonstrated that the binding sites of β-COP are different in TREK1 and TREK2. While β-COP binds to the N-terminus of TREK1 [18], it binds to the RRR motif in the C2 region of TREK2. In addition, β-COP increases the surface expression of TREK1 but restricts that of TREK2.

A previous study reported that the binding site for AKAP150 (amino acids V298 to R313) in TREK1 is conserved in the C1 region of TREK2, and AKAP150 increases the TREK2-mediated currents [16]. Since the AKAP150 binding region of TREK2 is not far from the RRR motif (β-COP binding site) of TREK2 (Appendix A), it seems to be worth examining the cooperative roles of AKAP150 and β-COP in the surface expression of TREK2 in future studies. Mtap2 is also reportedly associated with the C-terminus of TREK1 and TREK2 [17]. However, the Mtap2 binding site in the C-terminus of TREK1 (amino acids E335 to Q360) is partially conserved in the C-terminus of TREK2 and may overlap with β-COP binding sites in the C-terminus of TREK2 (Appendix A). Therefore, binding of Mtap2 with the C-terminus of TREK2 could interfere with binding of β-COP with the C-terminus of TREK2. Since both AKAP150 and Mtap2 bind to the C-terminus of TREK2 [17] and increase surface expression of TREK2, the possibility of competition between these proteins and β-COP is an interesting topic for future research.

From the perspective of ion channels, both TREK1 and TREK2 are responsible for outward K^+^ currents [5]. However, the effect of β-COP on K^+^ current regulation in cells where TREK1 and TREK2 are equally highly expressed is unclear. The difference in the effect of β-COP on K^+^ current change between TREK1 and TREK2 may be determined by the relative expression levels of TREK1 and TREK2 in each cell. Furthermore, the affinity of β-COP for the N terminus of TREK1 [18] and the RRR motif at the C2 segment of TREK2 may play crucial roles in determining the differential responses of TREK1 and TREK2 to β-COP. Further investigation is necessary to determine how β-COP affects the K^+^ current in these cells, since β-COP is a ubiquitous gene. It is also interesting to note the role of β-COP in the surface expression of the TREK1/TREK2 heterodimers since TREK1 can make heterodimers with TREK2 [37]. The reactivity of the TREK1/TREK2 heterodimer to ruthenium red was intermediate between TREK1 and TREK2 [37]. Therefore, the potassium current-regulating effect of the heterodimer by β-COP in cells expressing high levels of TREK1 and TREK2 may be intermediate between that of TREK1 and TREK2.

TREK channels form heterodimers with various K2P channels and perform several physiological functions in various tissues [38,39,40]. Therefore, it is necessary to study how physiological phenomena mediated by TREK-containing heterodimers are regulated by β-COP. Recently, we reported that β-COP enhances TWIK1/TREK1 heterodimer-mediated passive conductance in astrocytes of the mouse hippocampus [29]. Since TREK2 can also form heterodimers with TRESK [40], and these channels are responsible for background potassium current in primary sensory neurons of the dorsal root and trigeminal ganglia [41], β-COP may be involved in the surface expression of the TREK2/TRESK heterodimer. Another paper reported that a 2bp frameshift mutation of TRESK (TRESK-MT) in sensory neurons binds directly to TREK1 and TREK2, increases neuronal excitability through inhibition of these channels, and induces a migraine-like phenotype [42,43]. Based on the present results, it is also an interesting topic to study whether the neuronal excitability of sensory neurons can be regulated by β-COP.

## 5. Conclusions

TREK1, TREK2, and TRAAK belong to the same subfamily and share structural and functional similarities; however, they exhibit different mechanisms of action when interacting with β-COP. β-COP binds to TREK1 at the N-terminus to increase membrane ex-pression, whereas it binds to the C-terminus of TREK2 to decrease membrane expression, and β-COP does not associate with TRAAK. Binding between β-COP and TREK2 occurs in a sequence location-specific manner that is not shared with TREK1, resulting in the relocation of TREK2 from the membrane to the Golgi apparatus, thereby greatly reducing its channel function. These results provide exciting opportunities for researchers interested in binding proteins as they offer numerous experimental designs. Furthermore, research on the differential association of channels within the same group with the same protein can help expand the diversity of regulatory mechanisms and act as a catalyst for new discoveries in the field of ion channel-binding proteins.

## Figures and Tables

**Figure 1 cells-12-01500-f001:**
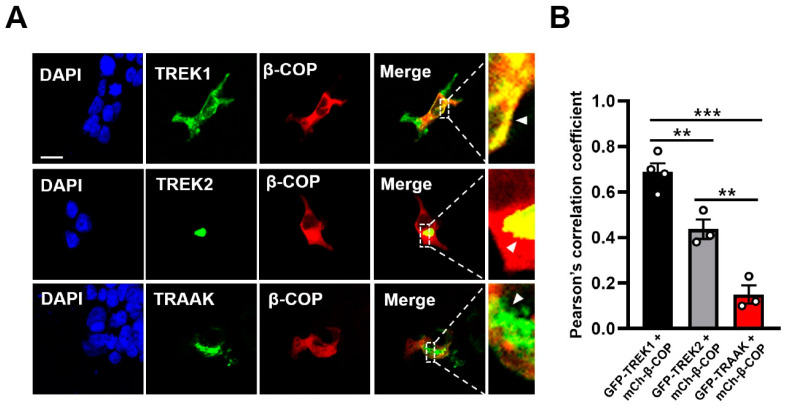
Association with β-COP differs for each member of the tandem of pore domains in a weak inward rectifying K^+^ channel (TWIK)-related K^+^ channel (TREK) family. (**A**) Representative images of HEK293T cells transfected with GFP-TREK1 and mCh-β-COP (up), GFP-TREK2 and mCh-β-COP (middle), or GFP-TRAAK and mCh-β-COP (low). Cell nuclei were stained with 4′,6-diamidino-2-phenylindole (DAPI). Yellow indicates the co-localized region. The white arrowheads in the enlarged image indicate that β-COP associates with TREK1 and TREK2, but not with TRAAK. Scare bar, 20 μm. (**B**) The co-localization of the GFP-tagged TREK family and mCh-β-COP was quantified from individual 10 regions of a cell using the Pearson’s correlation coefficients. All values are the mean ± standard error of the mean (SEM). ** *p* < 0.01, *** *p* < 0.001. (**C**) A representative image of a bimolecular fluorescence complementation (BiFC) assay. VC-tagged TREK1 and TREK2 showed fluorescent signals when co-transfected with VN-tagged β-COP, but not TRAAK. mCh-con represents the overall morphology of cells. Scale bar, 20 μm. (**D**) Co-immunoprecipitation assay in HEK293T cells. After co-transfection of Flag-tagged TREK family members with GFP-β-COP, immunoprecipitation with Flag was performed and blotted with GFP.

**Figure 2 cells-12-01500-f002:**
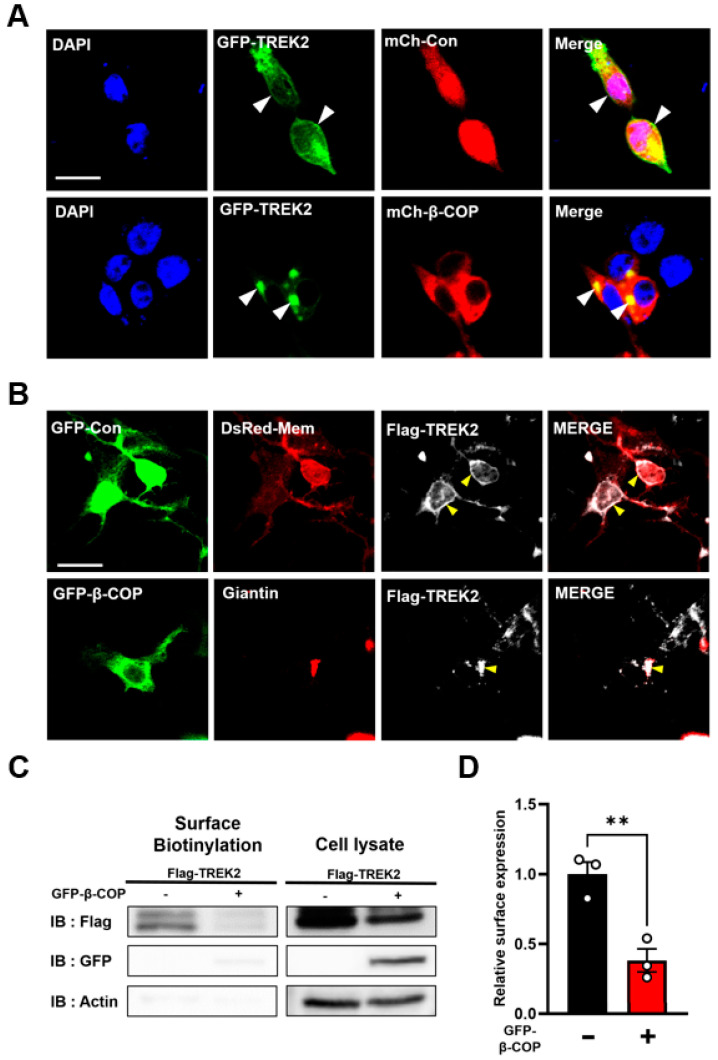
β-COP suppresses the membrane expression of TREK2. (**A**) Representative images of HEK293T cells transfected with GFP-TREK2, mCh-Con, and mCh-Con or mCh-β-COP. Cell nuclei were stained with DAPI. Yellow indicates the co-localized region. The white arrowheads in the enlarged image indicate the intracellular localization of TREK2 without or with mCh-β-COP. Scare bar, 20 μm. (**B**) Representative immunocytochemical images of HEK293T cells transfected with GFP-Con, DsRed-Mem, and Flag-TREK2 (up), GFP-β-COP, Giantin, and Flag-TREK2 (low). The cells were then stained with an anti-Flag antibody. The yellow arrowheads show the different localization of TREK2 in the presence or absence of β-COP. Scare bar, 20 μm. (**C**) Cell surface biotinylation experiments using HEK293T cells transfected with Flag-TREK2 with or without GFP-β-COP. (**D**) Normalized cell-surface expression values were quantified using three independent cell-surface biotinylation assays. All values are the mean ± standard error of the mean (SEM). ** *p* < 0.01. White circles indicate individual results. (**E**) Average I–V relationship derived from whole-cell currents in HEK293T cells transfected with GFP-TREK2, mCh-Con, or mCh-β-COP. (**F**) Normalized currents at +50 mV in (**E**). All values are the mean ± standard error of the mean (SEM). ** *p* < 0.01. White circles indicate individual currents.

**Figure 3 cells-12-01500-f003:**
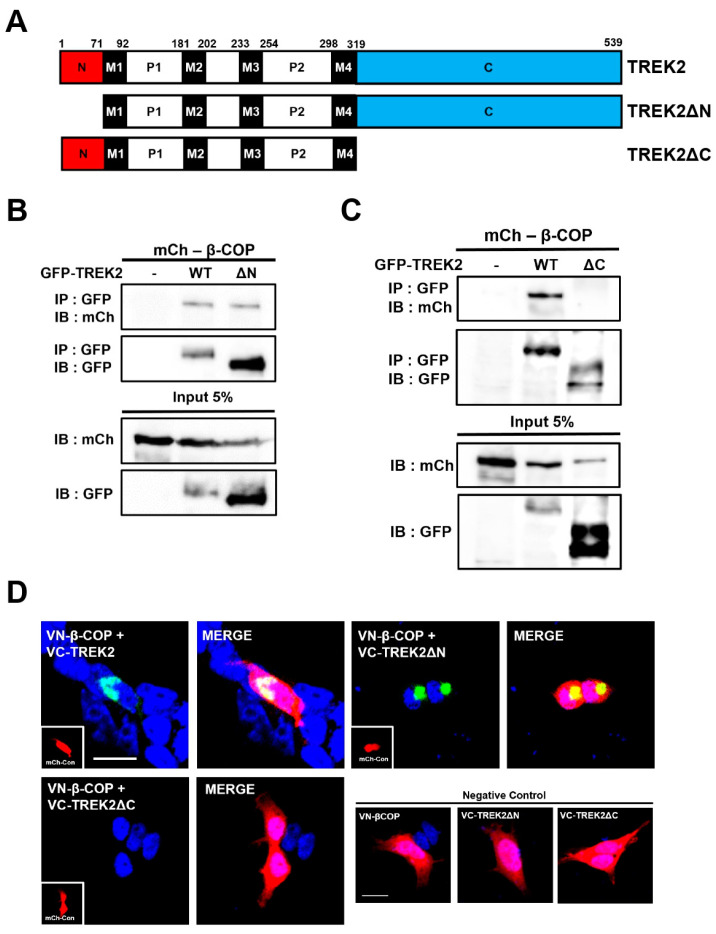
The C-terminus of TREK2 is a pivotal binding region for β-COP. (**A**) Schematic diagram of the sequences of TREK2, TREK2 N-terminal deletion mutant (TREK2ΔN), and TREK2 C-terminus deletion mutant (TREK2ΔC) used in the experiment. (**B**) Co-Immunopricipitation (Co-IP) in HEK293T cells. After transfection of GFP-tagged TREK2 or TREK2ΔN and mCh-β-COP, immunoprecipitation was carried out with GFP and blotted with mCh. (**C**) Co-IP in a HEK293T cell. After transfection of GFP-tagged TREK2 or TREK2ΔC and mCh-β-COP, immunoprecipitation was carried out with GFP and blotted with mCh. (**D**) Representative image of the BiFC assay. VN-tagged β-COP showed strong fluorescence when co-transfected with VC-tagged TREK2ΔN but not VC-tagged TREK2ΔC. mCh-Con represents the overall morphology of cells. Scale bar, 20 μm.

**Figure 4 cells-12-01500-f004:**
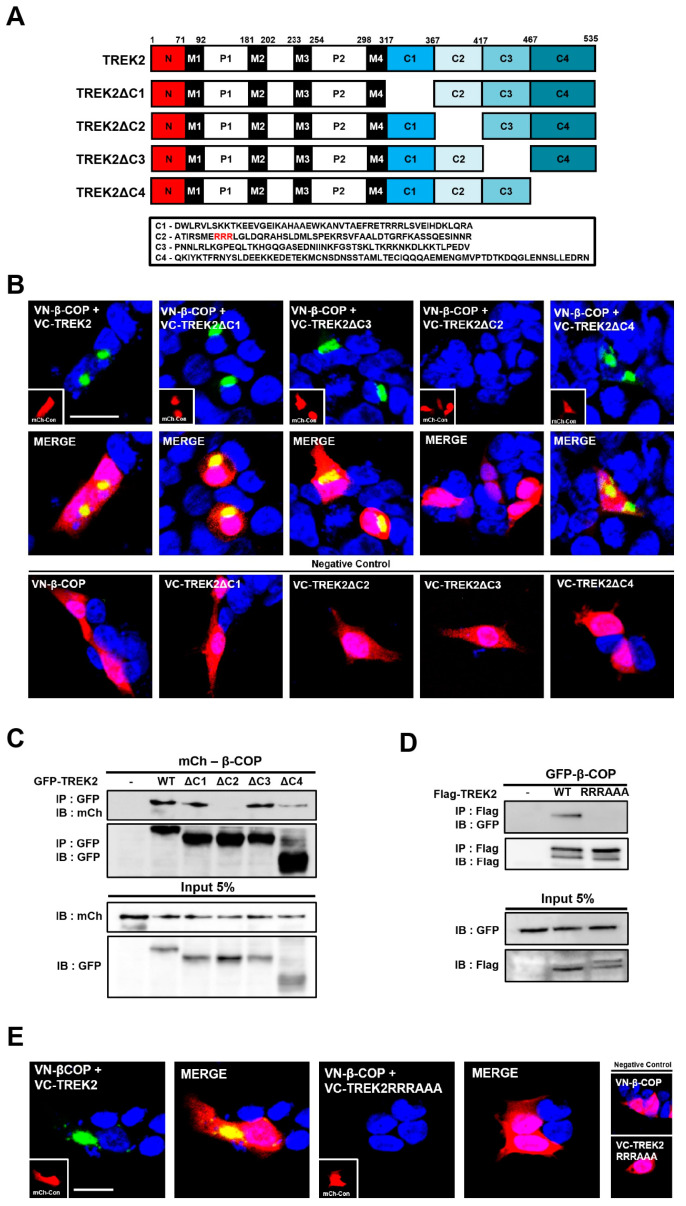
RRR motif in TREK2 is essential for binding to β-COP. (**A**) A schematic diagram of TREK2 and TREK2 C-terminus deletion mutants (from the top, TREK2ΔC1, TREK2ΔC2, TREK2ΔC3, TREK2ΔC4) used in the experiment. (**B**) A representative image of the BiFC experiment. VC-tagged TREK2 C-terminus series deletion mutants were each co-transfected with VN-tagged β-COP. mCh-Con represents the overall morphology of cells. Scale bar, 20 μm. (**C**) Co-IP assay in HEK293T cells. After co-transfection of each GFP-tagged TREK2 C-terminus series deletion mutant and mCh-β-COP, immunoprecipitation with GFP was performed and blotted with RFP. (**D**) Co-IP assay in HEK293T cells. After co-transfection of Flag-tagged TREK2 or TREK2RRRAAA with GFP-β-COP, immunoprecipitation with Flag was performed and blotted with GFP. (**E**) Representative image of the BiFC assay. No fluorescence signal was observed when VC-tagged TREK2RRRAAA was co-transfected with VN-tagged β-COP. mCh-Con represents the overall cellular morphology. Scale bar, 20 μm.

**Figure 5 cells-12-01500-f005:**
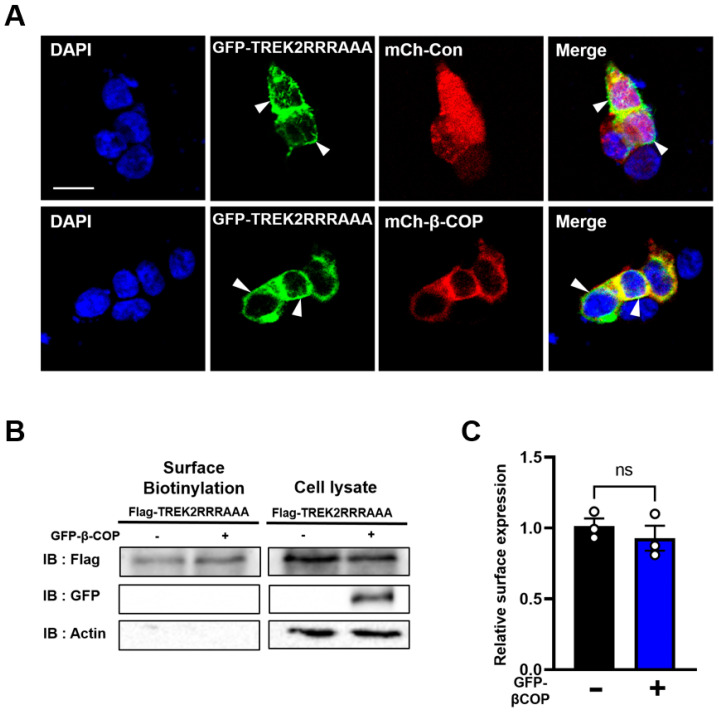
β-COP does not affect the membrane expression of the TREK2RRRAAA mutant. (**A**) Representative images of HEK293T cells transfected with GFP-TREK2RRRAAA, mCh-Con, and mCh-Con or mCh-β-COP. Cell nuclei were stained with DAPI. White arrowheads indicate membrane localization of TREK2RRRAAA is not changed in the presence of mCh-β−COP. Scare bar, 20 μm. (**B**) Cell surface biotinylation experiments with HEK293T cells transfected with Flag-TREK2RRRAAA, with or without GFP-β-COP. (**C**) Normalized cell surface expression values were quantified using three independent biotinylation assays. All values are the mean ± standard error of the mean (SEM). ns: not significant. White circles indicate individual raw data. (**D**) Average I–V relationship derived from whole-cell currents in HEK293T cells transfected with GFP-TREK2RRRAAA and mCh-Con or mCh-β-COP. (**E**) Normalized currents at +50 mV in (**D**). All values are the mean ± standard error of the mean (SEM). ns: not significant. White circles indicate individual currents.

## Data Availability

All data will be available upon reasonable request by emailing the corresponding author.

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
