# Peer review of "β-COP Suppresses the Surface Expression of the TREK2"

_cells, 2023, doi:10.3390/cells12111500_

Round 1
Reviewer 1 Report
The manuscript submitted by SEONG-SEOP Kim and collaborators entitled “β-COP suppresses the surface expression of the TREK-2” reports the putative role of β-COP a subunit of COP1 protein in the regulation of membrane TREK-2 K2P potassium channel expression. β-COP is a vesicular binding protein and in their studies, authors examined how this protein regulates the TREK-2 channel expression in cell membrane. The stemming results of the study are really interesting and relevant, but it misses however, several things to clarify the role β-COP protein. The major and minor points are the following:
Major comments:
- In their β-COP Co expression experiments figure 1, why authors use arachidonic acid to activate TRAAK channel only? TRAAK channel is responsible of basal current which is sufficiently without activation by PUFAs as AA…
- In their study, authors use patch clamp experiments to record channel in different conditions of expression dependent of β-COP Co expression. The whole cell configuration of the patch clamp was used for the recordings and the currents were expressed in amplitude pA. Why authors did not express currents in density pA/pF because is the best manner to mean current amplitudes independently of the high of the patched cells?
- In their study, authors use patch clamp experiments to record channel in different conditions of expression dependent of β-COP Co expression using the whole cell configuration. In these experimental conditions the intracellular medium or pipette medium has 322 mosmol/l of osmolarity and the extracellular medium or bath medium has 350 mosmol/l of osmolarity. These values are too big because basically osmolarity is around 295 mosmol/l for the intracellular medium and 305 mosmol/l in the extracellular medium with 10 mosmol/l of difference between both. How authors could explain these conditions which is not really good for the cells and the patch with impact on cell volume and swelling currents?
Minor comments:
- Line 34: the sentence is too focus on K2P channels which are not the only type of channels.
- Typing error in several sentences of the manuscript with cutting different words by - .
Author Response
Responses are in bold
Changed parts in the revised manuscript were marked in yellow.
Reviewer 1.
The manuscript submitted by SEONG-SEOP Kim and collaborators entitled “β-COP suppresses the surface expression of the TREK-2” reports the putative role of β-COP a subunit of COP1 protein in the regulation of membrane TREK-2 K2P potassium channel expression. β-COP is a vesicular binding protein and in their studies, authors examined how this protein regulates the TREK-2 channel expression in cell membrane. The stemming results of the study are really interesting and relevant, but it misses however, several things to clarify the role β-COP protein. The major and minor points are the following:
We thank the reviewer for the positive and encouraging comments.
Major comments:
-In their β-COP Co expression experiments figure 1, why authors use arachidonic acid to activate TRAAK channel only? TRAAK channel is responsible of basal current which is sufficiently without activation by PUFAs as AA…
We thank the reviewer’s comments. However, we did not use arachidonic acid to activate TRAAK in Figure 1, nor did we used it in any of the experiments in this paper. To avoid any confusion, we have updated the legend of Figure1 by replacing ‘GFP-TWIK-related acid-arachidonic activated K+ (TRAAK)’ with ‘GFP-TRAAK’ in the yellow-colored part of line 236.
-In their study, authors use patch clamp experiments to record channel in different conditions of expression dependent of β-COP Co expression. The whole cell configuration of the patch clamp was used for the recordings and the currents were expressed in amplitude pA. Why authors did not express currents in density pA/pF because is the best manner to mean current amplitudes independently of the high of the patched cells?
According to reviewer’s comments, we changed our electrophysiological data with pA/pF in Figure 2E and Figure 5D in our revised manuscript.
-In their study, authors use patch clamp experiments to record channel in different conditions of expression dependent of β-COP Co expression using the whole cell configuration. In these experimental conditions the intracellular medium or pipette medium has 322 mosmol/l of osmolarity and the extracellular medium or bath medium has 350 mosmol/l of osmolarity. These values are too big because basically osmolarity is around 295 mosmol/l for the intracellular medium and 305 mosmol/l in the extracellular medium with 10 mosmol/l of difference between both. How authors could explain these conditions which is not really good for the cells and the patch with impact on cell volume and swelling currents?
We thank the reviewer’s comments. We routinely used the same bath solution and pipette solution in our previously reported papers (Hwang et al., Nature Communications 2012; Bae et al., Cells 2020; Kim et al., Cells 2022). When we prepare the solutions, the osmotic pressure of solutions was measured with OSMETTE (5004 MICRO-OSMETTE Automatic High Sensitivity 50µL Osmometer, PRECISION SYSTEMS INC.) and we did not find the prominent changes in cell volume and whole-cell currents in control cells. Nevertheless, we could not exclude the possibility that these solutions affect the volume-regulated currents. However, we thought that the osmolarity of used solutions could not affect TREK-2 mediated currents in our manuscript, since we used the same experimental condition to measure the effects of β-COP on the TREK-2 currents.
Minor comments:
-Line 34: the sentence is too focus on K2P channels which are not the only type of channels.
According to reviewer’s comments, we rewrote introduction section of our revised manuscript (line 34-35). ‘Potassium homeostasis is mediated by the two-pore domain K+ (K2P) channels, which have leaky characteristics’ is replaced by ‘Two pore domain K+ (K2P) channels have leaky properties and have been reported to mediate cellular potassium homeostasis.’.
-Typing error in several sentences of the manuscript with cutting different words by - .
We apologize about our careless mistakes. The typos have been corrected in our revised manuscript.
Reviewer 2 Report
The manuscript of Kim et al. entitled: “β-COP suppresses the surface expression of the TREK2” describes a regulation role of β-COP binding to the TREK2 channel. Using multiple experimental methods, the authors document that interaction of TWIK-related K+ channel 2 (TREK2) with a transporting vesicle-associated coat protein, β-COP, decreased a channel surface expression in HEK-293T cells. Furthermore, molecular mechanisms underlying β-COP/TREK2 interaction were studied. The authors show that the interaction region is located within the C-terminus of the TREK2 channel. More specifically, the second RRR motif within the C2 segment of the TREK2 C-terminus seems to be critical for β-COP binding and regulation of TREK2 surface expression. Although, the manuscript is sufficiently promising, I have identified some concerns that need to be addressed by the authors.
Major concerns:
1. Introduction: Please briefly explain abbreviations such as TASK, TRESK, AKAP150 and Mtap2 in order to attract more readers of the manuscript.
2. Results, Figs. 1C and 2C: Why are there more bands for the Flag-TREK2 channel? It was not seen for GPT-TREK2 in Fig. 3B. On the other hand, for the truncated GFP-TREK2 missing the C-terminus also two bands were stained (Fig. 3C). This pattern should be explained.
3. Results, Fig. 1A: The authors stated that TREK1 and β-COP co-localized throughout the cell, particularly near the cell membrane. However, only one cell shown in Fig. 1A (top panel) supports this conclusion. In the other one, yellow color is spread throughout the cell and plasma membrane seems to be green, the color related to the TREK1 signal. In addition, it was concluded that the co-localization of the TREK2 channel and β-COP was localized in a Golgi-like organelle adjacent to the nucleus (Fig. 1A, middle panel). However, on the base of Golgi staining in Fig. 2B (bottom panel), it seems to me that the Golgi system is not so large as was interpreted in Fig. 1A. The authors should provide more solid representative images of HEK293T cells.
4. Results, Fig. 4C: The authors stated that RRR motif within the C2 segment of the TREK2 C-terminus is critical for β-COP binding. It was proved by replacing this motif by AAA sequence. According to me the C4 segment might be also involved in such interaction because β-COP binding was substantially decreased when the C4 segment was missing. This issue should be discussed.
Minor concerns:
1. Abstract, Line 17: TREK1 and TREK2 belong to the TREK subfamily and not the TREK1 subfamily. Please correct.
2. Results, Figs. 2D and 5C: Labeling of Y-axis is missing.
The English language only needs minor editing to correct some typos. Additionally, many words were hyphenated even though they were not at the end of a line. Such hyphenation should be removed.
Author Response
Responses are in bold
Changed parts in the revised manuscript were marked in yellow.
Reviewer 2
The manuscript of Kim et al. entitled: “β-COP suppresses the surface expression of the TREK2” describes a regulation role of β-COP binding to the TREK2 channel. Using multiple experimental methods, the authors document that interaction of TWIK-related K+ channel 2 (TREK2) with a transporting vesicle-associated coat protein, β-COP, decreased a channel surface expression in HEK-293T cells. Furthermore, molecular mechanisms underlying β-COP/TREK2 interaction were studied. The authors show that the interaction region is located within the C-terminus of the TREK2 channel. More specifically, the second RRR motif within the C2 segment of the TREK2 C-terminus seems to be critical for β-COP binding and regulation of TREK2 surface expression. Although, the manuscript is sufficiently promising, I have identified some concerns that need to be addressed by the authors.
We thank the reviewer’s helpful comments, which are helpful to make our paper more compelling.
Major concerns:
- Introduction: Please briefly explain abbreviations such as TASK, TRESK, AKAP150 and Mtap2 in order to attract more readers of the manuscript.
According to reviewer’s comments, we explained abbreviations in the introduction of our revised manuscript (the yellow-colored part, line 43~50).
- Results, Figs. 1C and 2C: Why are there more bands for the Flag-TREK2 channel? It was not seen for GPT-TREK2 in Fig. 3B. On the other hand, for the truncated GFP-TREK2 missing the C-terminus also two bands were stained (Fig. 3C). This pattern should be explained.
It has been known that TREK1 and TREK2 were N-glycosylation, a post-translational modification (Wiedmann, Felix et al., 2019; Hyoweon Bang et al., 2000). Indeed, we also previously confirmed glycosylation of TREK channels by treatment of N-glycosidase (data not shown). In addition, depending on the size of tagged GFP or Flag, glycosylated band and non-glycosylated band of channel proteins are not clearly separated in the gel. We mentioned about the existence of glycosylated ion channels in Western blot data (the yellow-colored part, line 228).
- Results, Fig. 1A: The authors stated that TREK1 and β-COP co-localized throughout the cell, particularly near the cell membrane. However, only one cell shown in Fig. 1A (top panel) supports this conclusion. In the other one, yellow color is spread throughout the cell and plasma membrane seems to be green, the color related to the TREK1 signal. In addition, it was concluded that the co-localization of the TREK2 channel and β-COP was localized in a Golgi-like organelle adjacent to the nucleus (Fig. 1A, middle panel). However, on the base of Golgi staining in Fig. 2B (bottom panel), it seems to me that the Golgi system is not so large as was interpreted in Fig. 1A. The authors should provide more solid representative images of HEK293T cells.
According to reviewer’s helpful comments, we replaced Figure 1 with new one.
- Results, Fig. 4C: The authors stated that RRR motif within the C2 segment of the TREK2 C-terminus is critical for β-COP binding. It was proved by replacing this motif by AAA sequence. According to me the C4 segment might be also involved in such interaction because β-COP binding was substantially decreased when the C4 segment was missing. This issue should be discussed.
We appreciate reviewer’s appropriate comments. In Co-IP data, TREK2 ΔC4 has a weak band compared to other deletion mutants. In general, β-COP is known to recognize and bind to highly conserved sequences (di-lysine (KxK), di-arginine (RxR), KDEL) through several papers Letourneur, F et al. 1994; Bikard, Y et al. 2019; Étienne St-Louis et al. 2016). Since the RRR sequence is existed at the C2 region of TREK2, we examined whether the RRR sequence of TREK2 is critical for β-COP binding in our manuscript. We agree with reviewer’s opinion and cannot exclude a possibility that β-COP also binds with C4 region of TREK2 by recognizing a non-conserved sequence. We discussed this possibility in our revised manuscript (line 450-456).
Minor concerns:
- Abstract, Line 17: TREK1 and TREK2 belong to the TREK subfamily and not the TREK1 subfamily. Please correct.
We apologize about our careless mistake. The typo error is corrected in our revised manuscript (yellow colored part in line 17).
- Results, Figs. 2D and 5C: Labeling of Y-axis is missing.
We apologize about our careless mistake. Figures 2D and 5C are replaced by new ones in our revised manuscript.
The English language only needs minor editing to correct some typos. Additionally, many words were hyphenated even though they were not at the end of a line. Such hyphenation should be removed.
We apologize about our careless mistake. It seems that many words were hyphenated in the process of transferring our original draft to the template of the Cells journal. We corrected these words in our revised manuscript.
Reviewer 3 Report
The manuscript by Kim et al. describes the interaction between beta-COP and TREK-2. The Authors found that in contrast with TREK-1, beta-COP suppresses the expression of TREK-2 via sequestering it in the secretory compartment of the cell. In general, the manuscript is well written, there are only a few stylistical and typographical errors, but nothing which can not be corrected with a thorough rereading of the text. The experiments are mostly well described and performed, although there are some points which should be paid attention to. A major weakness of the manuscript is the discussion, where the results are not sufficiently placed into the context of the field. Furthermore, important papers (relating to heteromeric assembly of TREK-2 subunits with TREK-1 or TRESK) are completely omitted from discussion which significantly weakens the manuscript in its current form.
Major questions:
-Figure 1A: Was the degree of colocalization between beta-COP and the different TREK subunits quantified? What about the subcellular localization of the channel subunits?
-Figure 1B: Bimolecular fluorescence complementation is a method where adequate controls are very important. Where the experiments performed with opposite tag orientation (eg. VC-BetaCOP and VN-TREK-1 to see if tag position influences the signal)? Furthermore, is there a positive control for the VC-TRAAK construct?
-Figure 1B: Was there any quantification performed for the BiFC assay?
-Figure 3C: Was there a positive control for the TREK-2deltaC construct in the BiFC assay?
-Figure 4: Was the functionality of the different TREK-2 C-terminal deletion constructs and the RRR ïƒ AAA mutant confirmed? Where there any positive controls for these mutants in the BiFC assay?
Discussion: This is a major weakness of the paper and most be improved significantly before publication.
-The phosphorylation sites on the C-terminal (S300, E306, S333, or S351) are present in both TREK-1 and TREK-2. Presence of these in TREK-1 is not sufficient to explain why it does not bind to the C-terminal of TREK-1
-Biological relevance: TREK-1 and TREK-2 are highly expressed in the peripheral sensory neurons. What would the relevance of beta-COP be in these neurons.
-“Unlike TREK1, which functions as a heterodimer of TWIK1” : This statement in this form is false and this part requires extensive rewriting. TREK-1 has important physiological functions as a homodimer in a variety of tissues (for recent reviews see PMID: PMC8396510 and PMID: PMC6470294). Furthermore, the fact that TREK-1 has been shown to form heterodimers with TREK-2 in expression systems and native neurons by multiple groups and also with TRAAK and TRESK in expression systems has been completely omitted from the discussion ( PMC4839434, PMC4839437, PMID: 30573346, PMC4919449 and PMC7458809). These results should be discussed in the context of the interaction between Beta-COP and TREK channels, especially given the fact that this protein influences TREK-1 and TREK-2 in opposite directions.
-“The physiological K2P channel partner of TREK2 has not yet been identified” : This statement is not true. Beyond the earlier work showing the importance of homoeric TREK-2 in sensory neurons (such as PMC5901755 and PMID: 16495368) recent work has shown that TREK-2 can heterodimerize with TRESK in primary sensory neurons (see previous references) and disruption of this interaction leads to increased neuronal sensitivity, hyperalgesia and migraine pain (PMC8379698 and PMC9977718).
Minor comments:
-The ‘+’ in K+ is often not placed in superscript throughout the text. Please correct.
-Line 44: The regulation of TRESK by 14-3-3 is more complex than merely influencing the plasma membrane expression as suggested by the sentence. Please rephrase.
-Methods: Details of transfection (how much DNA, how long after transfection were the cells used for experiments) are missing
-Line 146: What antibody was used?
-Figure 2D: The y-axis does not have a label.
-Figure 2E: The plot suggests that steady-state currents were measured at the end of voltage steps. However, the methods section states that a ramp protocol was used to measure currents. Why aren’t the original ramp currents shown then?
-Figure 2E and 2F: The figure legend suggests that these are two separate sets of experiments. Is this correct? If yes, then what were the sample sizes used to calculate SEM values in Figure 2E?
-Figure 5C: The y-axis does not have a label.
-Figure 5D: The plot suggests that steady-state currents were measured at the end of voltage steps. However, the methods section states that a ramp protocol was used to measure currents. Why aren’t the original ramp currents shown then?
-Figure 5D and 5E: The figure legend suggests that these are two separate sets of experiments. Is this correct? If yes, then what were the sample sizes used to calculate SEM values in Figure 5D?
A thorough proofreading of the text would help catch minor typing and grammatical errors.
Author Response
Responses are in bold
Changed parts in the revised manuscript were marked in yellow.
Reviewer 3
The manuscript by Kim et al. describes the interaction between beta-COP and TREK-2. The Authors found that in contrast with TREK-1, beta-COP suppresses the expression of TREK-2 via sequestering it in the secretory compartment of the cell. In general, the manuscript is well written, there are only a few stylistical and typographical errors, but nothing which cannot be corrected with a thorough rereading of the text. The experiments are mostly well described and performed, although there are some points which should be paid attention to. A major weakness of the manuscript is the discussion, where the results are not sufficiently placed into the context of the field. Furthermore, important papers (relating to heteromeric assembly of TREK-2 subunits with TREK-1 or TRESK) are completely omitted from discussion which significantly weakens the manuscript in its current form.
Major questions:
-Figure 1A: Was the degree of colocalization between beta-COP and the different TREK subunits quantified? What about the subcellular localization of the channel subunits?
We appreciate reviewer’s appropriate comments. According to reviewer’s suggestion, the Pearson correlation coefficients were determined using a software of Nikon A1 confocal microscope to quantify co-localization of the merged images. We exchanged Figure 1 with new figure and edited figure legend (Page 29 and 31). We also added description for quantification method in section of Materials and Methods (line 138-141). Changes in revised manuscript were indicated by yellow color (line 209-211). To specify the intracellular location of TREK2, which is of interest in this paper, cell membrane and Golgi markers were used (Figure 2B). TREK2 is expressed in the cell membrane under normal conditions; however, in the presence of additional β-COP, the intracellular expression region of TREK2 changes to the Golgi apparatus.
-Figure 1B: Bimolecular fluorescence complementation is a method where adequate controls are very important. Where the experiments performed with opposite tag orientation (eg. VC-BetaCOP and VN-TREK-1 to see if tag position influences the signal)? Furthermore, is there a positive control for the VC-TRAAK construct?
Thanks for reviewer’s helpful comments. We initially found that strong BiFC signals are generated from the homodimerization of TREK-1, TREK-2 and TRAAK, respectively. In addition, since the BiFC signal between TREK-1 and β-COP have been reported in the previous paper (Kim et al., BBRC 2010; Kim et al., Cells 2022), BiFC signals between TREK-1 and β-COP can be used as a positive control in this manuscript. We already mentioned previous reported BiFC signals between TREK-1 and β-COP in the results section (line 217-218).
-Figure 1B: Was there any quantification performed for the BiFC assay?
Thanks for reviewer’s comments. However, the BiFC assay is an experimental technique to measure the protein-protein interaction using a pair of split Venus fragments composed of VN and VC. In general, BiFC assay is shown as 'with fluorescence' or 'without fluorescence'. To examine the quantification of protein-protein interaction, we performed immunoprecipitation (IP) experiments (Figure 1D).
-Figure 3C: Was there a positive control for the TREK-2deltaC construct in the BiFC assay?
BiFC signal between TREK-2 and β-COP is a positive control. We added this positive control in Figure 3D in our revised manuscript.
-Figure 4: Was the functionality of the different TREK-2 C-terminal deletion constructs and the RRR à AAA mutant confirmed? Where there any positive controls for these mutants in the BiFC assay?
Thanks for reviewer’s questions. As shown in Figures (1C, 2C, 3B-C, 4C-D, 5B), cellular imaging and Western blot experiments showed protein expression of TREK-2 mutants. In addition, the TREK2 RRRAAA mutant displayed huge currents (Figure 5D-E). Therefore, we believe that TREK-2 mutants functionally expressed.
BiFC signal between TREK-2 and β-COP can be used as a positive control. We added BiFC signal between TREK-2 and β-COP as a positive control in Figure 1, 3 and 4.
Discussion: This is a major weakness of the paper and most be improved significantly before publication.
-The phosphorylation sites on the C-terminal (S300, E306, S333, or S351) are present in both TREK-1 and TREK-2. Presence of these in TREK-1 is not sufficient to explain why it does not bind to the C-terminal of TREK-1
Thanks for reviewer’s proper comments. We apologize about our careless mistake. Although we know the the S300, E306, S333, or S351 equally conserve in the C-terminus of TREK1 and TREK2, we are confused while writing other papers. As shown in Supplementary Figure 1, RRR motif in C2 region of TREK2 is unique to TREK2. We rewrote the section of discussion (line 434-438).
-Biological relevance: TREK-1 and TREK-2 are highly expressed in the peripheral sensory neurons. What would the relevance of beta-COP be in these neurons.
Thanks for reviewer’s comments. According to reviewer’s comments, we rewrote the the section of discussion (line 459-471).
-“Unlike TREK1, which functions as a heterodimer of TWIK1” : This statement in this form is false and this part requires extensive rewriting. TREK-1 has important physiological functions as a homodimer in a variety of tissues (for recent reviews see PMID: PMC8396510 and PMID: PMC6470294). Furthermore, the fact that TREK-1 has been shown to form heterodimers with TREK-2 in expression systems and native neurons by multiple groups and also with TRAAK and TRESK in expression systems has been completely omitted from the discussion ( PMC4839434, PMC4839437, PMID: 30573346, PMC4919449 and PMC7458809). These results should be discussed in the context of the interaction between Beta-COP and TREK channels, especially given the fact that this protein influences TREK-1 and TREK-2 in opposite directions.
According to reviewer’s comments, we rewrote the section of discussion (line 459-471).
-“The physiological K2P channel partner of TREK2 has not yet been identified” : This statement is not true. Beyond the earlier work showing the importance of homoeric TREK-2 in sensory neurons (such as PMC5901755 and PMID: 16495368) recent work has shown that TREK-2 can heterodimerize with TRESK in primary sensory neurons (see previous references) and disruption of this interaction leads to increased neuronal sensitivity, hyperalgesia and migraine pain (PMC8379698 and PMC9977718).
We also rewrote the section of discussion in our revised manuscript (line 472-484).
Minor comments:
-The ‘+’ in K+ is often not placed in superscript throughout the text. Please correct.
We apologize about our careless mistake. We corrected them in our revised manuscript.
-Line 44: The regulation of TRESK by 14-3-3 is more complex than merely influencing the plasma membrane expression as suggested by the sentence. Please rephrase.
We rephrase this sentence (line 44-46).
-Methods: Details of transfection (how much DNA, how long after transfection were the cells used for experiments) are missing
We rewrote the methods for transfection (line 115~117).
-Line 146: What antibody was used?
Anti-GFP antibody was used in this experiment (line 150).
-Figure 2D: The y-axis does not have a label.
We apologize about our careless mistake. Figure 2D is replaced by new one in our revised manuscript.
-Figure 2E: The plot suggests that steady-state currents were measured at the end of voltage steps. However, the methods section states that a ramp protocol was used to measure currents. Why aren’t the original ramp currents shown then?
According to reviewer’s request, Figure 2E is displayed by original ramp currents in our revised manuscript.
-Figure 2E and 2F: The figure legend suggests that these are two separate sets of experiments. Is this correct? If yes, then what were the sample sizes used to calculate SEM values in Figure 2E?
Figures 2E and 2F are obtained from same experiments. We apologize about our confusing description. To make clear it, we rewrote the legend of the figure.
-Figure 5C: The y-axis does not have a label.
Thanks for your careful comment. Figure 5C is replaced by new one in our revised manuscript.
-Figure 5D: The plot suggests that steady-state currents were measured at the end of voltage steps. However, the methods section states that a ramp protocol was used to measure currents. Why aren’t the original ramp currents shown then?
According to reviewer’s request, Figure 5D is also displayed by original ramp currents in our revised manuscript.
-Figure 5D and 5E: The figure legend suggests that these are two separate sets of experiments. Is this correct? If yes, then what were the sample sizes used to calculate SEM values in Figure 5D?
We apologize about our confusing description. Figures 5D and 5E are also obtained from same experiments. To make clear it, we rewrote the legend of the figure.
A thorough proofreading of the text would help catch minor typing and grammatical errors.
We apologize about our careless mistake. It seems that many words were hyphenated in the process of transferring our original draft to the template provided by Cells journal. We corrected these words in our revised manuscript.
Round 2
Reviewer 1 Report
Responses are in bold
Changed parts in the revised manuscript were marked in yellow.
Reviewer 1.
The manuscript submitted by SEONG-SEOP Kim and collaborators entitled “β-COP suppresses the surface expression of the TREK-2” reports the putative role of β-COP a subunit of COP1 protein in the regulation of membrane TREK-2 K2P potassium channel expression. β-COP is a vesicular binding protein and in their studies, authors examined how this protein regulates the TREK-2 channel expression in cell membrane. The stemming results of the study are really interesting and relevant, but it misses however, several things to clarify the role β-COP protein. The major and minor points are the following:
We thank the reviewer for the positive and encouraging comments.
Major comments:
-In their β-COP Co expression experiments figure 1, why authors use arachidonic acid to activate TRAAK channel only? TRAAK channel is responsible of basal current which is sufficiently without activation by PUFAs as AA…
We thank the reviewer’s comments. However, we did not use arachidonic acid to activate TRAAK in Figure 1, nor did we used it in any of the experiments in this paper. To avoid any confusion, we have updated the legend of Figure1 by replacing ‘GFP-TWIK-related acid-arachidonic activated K+ (TRAAK)’ with ‘GFP-TRAAK’ in the yellow-colored part of line 236.
OK with this answer and the modifications
-In their study, authors use patch clamp experiments to record channel in different conditions of expression dependent of β-COP Co expression. The whole cell configuration of the patch clamp was used for the recordings and the currents were expressed in amplitude pA. Why authors did not express currents in density pA/pF because is the best manner to mean current amplitudes independently of the high of the patched cells?
According to reviewer’s comments, we changed our electrophysiological data with pA/pF in Figure 2E and Figure 5D in our revised manuscript.
OK with this answer and the modifications
-In their study, authors use patch clamp experiments to record channel in different conditions of expression dependent of β-COP Co expression using the whole cell configuration. In these experimental conditions the intracellular medium or pipette medium has 322 mosmol/l of osmolarity and the extracellular medium or bath medium has 350 mosmol/l of osmolarity. These values are too big because basically osmolarity is around 295 mosmol/l for the intracellular medium and 305 mosmol/l in the extracellular medium with 10 mosmol/l of difference between both. How authors could explain these conditions which is not really good for the cells and the patch with impact on cell volume and swelling currents?
We thank the reviewer’s comments. We routinely used the same bath solution and pipette solution in our previously reported papers (Hwang et al., Nature Communications 2012; Bae et al., Cells 2020; Kim et al., Cells 2022). When we prepare the solutions, the osmotic pressure of solutions was measured with OSMETTE (5004 MICRO-OSMETTE Automatic High Sensitivity 50µL Osmometer, PRECISION SYSTEMS INC.) and we did not find the prominent changes in cell volume and whole-cell currents in control cells. Nevertheless, we could not exclude the possibility that these solutions affect the volume-regulated currents. However, we thought that the osmolarity of used solutions could not affect TREK-2 mediated currents in our manuscript, since we used the same experimental condition to measure the effects of β-COP on the TREK-2 currents.
Authors have to be careful with this big difference between intra and extra cellular osmolarity!! Basically, physiological osmolarity for all of cell type (lines and primary cells) is 295 mosmol/L in intracellular medium and 300-305 for extracellular medium with 10 mosmol/l maximum of difference between both…
If they use other values and differences it’s not the best and physiological conditions…
Minor comments:
-Line 34: the sentence is too focus on K2P channels which are not the only type of channels.
According to reviewer’s comments, we rewrote introduction section of our revised manuscript (line 34-35). ‘Potassium homeostasis is mediated by the two-pore domain K+ (K2P) channels, which have leaky characteristics’ is replaced by ‘Two pore domain K+ (K2P) channels have leaky properties and have been reported to mediate cellular potassium homeostasis.’.
OK with this answer and the modifications
-Typing error in several sentences of the manuscript with cutting different words by - .
We apologize about our careless mistakes. The typos have been corrected in our revised manuscript.
OK with this answer and the modifications
Reviewer 2 Report
I am satisfied with response of the authors. All my concerns were appropriately addressed. The manuscript is now ready for publishing.
Reviewer 3 Report
The manuscript has been much improved. I recommend publication.